# Characterizing Billbug (*Sphenophorus* spp.) Seasonal Biology Using DNA Barcodes and a Simple Morphometric Analysis

**DOI:** 10.3390/insects12100930

**Published:** 2021-10-13

**Authors:** Marian M. Rodriguez-Soto, Douglas S. Richmond, Ricardo A. Ramirez, Xi Xiong, Laramy S. Enders

**Affiliations:** 1Department of Entomology, Purdue University, West Lafayette, IN 47907, USA; drichmond@purdue.edu (D.S.R.); lenders@purdue.edu (L.S.E.); 2Department of Biology, Utah State University, Logan, UT 84322, USA; ricardo.ramirez@usu.ed; 3Division of Plant Sciences, University of Missouri, Columbia, MO 65211, USA; xiongx@missouri.edu

**Keywords:** turfgrass, IPM, species complex, COI, 18S, ITS2

## Abstract

**Simple Summary:**

Billbugs (*Sphenophorus* spp.) are a group of grass-feeding weevils considered to be one of the most important and widespread insect pests of turfgrass. However, our limited understanding of regional variation in billbug species composition and inability to identify the damaging larval stage to species level, has hindered our ability to resolve the seasonal biology of many billbug species and constrained development of effective management approaches. In this study, we developed a robust DNA barcoding approach for identification of morphologically cryptic billbug larvae. Using this molecular tool combined with larval head capsule measurements we characterized regional variation in billbug species and developed larval phenology charts. Our approach provides researchers with the molecular tools necessary to fill critical gaps in our understanding of billbug seasonal biology and will facilitate the development of improved turfgrass pest management programs.

**Abstract:**

Billbugs (*Sphenophorus* spp.) are a complex of grass-feeding weevil species that reduce the aesthetic and functional qualities of turfgrass. Effective billbug monitoring and management programs rely on a clear understanding of their seasonal biology. However, our limited understanding of regional variation in the species compositions and seasonal biology of billbugs, stemming primarily from our inability to identify the damaging larval stage to species level, has hindered efforts to articulate efficient IPM strategies to growers. We used a combination of DNA barcoding methods and morphometric measures to begin filling critical gaps in our understanding of the seasonal biology of the billbug species complex across a broad geographic range. First, we developed a DNA barcoding reference library using cytochrome oxidase subunit 1 (COI) sequences from morphologically identified adult billbugs collected across Indiana, Missouri, Utah and Arizona. Next, we used our reference library for comparison and identification of unknown larval specimens collected across the growing season in Utah and Indiana. Finally, we combined our DNA barcoding approach with larval head capsule diameter, a proxy for developmental instar, to develop larval phenology charts. Adult COI sequences varied among billbug species, but variation was not influenced by geography, indicating that this locus alone was useful for resolving larval species identity. Overlaid with head capsule diameter data from specimens collected across the growing season, a better visualization of billbug species composition and seasonal biology emerged. This approach will provide researchers with the tools necessary to fill critical gaps in our understanding of billbug biology and facilitate the development of turfgrass pest management programs.

## 1. Introduction

In applied entomology, delimiting species complexes derived from different types of speciation (i.e., sympatric or cryptic) is fundamental to insect biological research [1,2,3]. Species complexes, composed of a group of closely related species, often lack morphological characters for species identification, which can lead to shortcomings in our understanding of a specific insect’s biology and impede the development of new strategies to manage pests [4,5,6,7,8,9]. Misidentification of pest species can reduce the effectiveness of management programs targeting plant and human diseases vectors [5,8], slow the advancement of biological control efforts [6] and potentially increase grower dependency on the prophylactic use of insecticides. Conversely, proper identification of pests that occur in species complexes can facilitate the understanding of pest biology [10] necessary to optimize management efforts [4] and allow growers to implement more judicious insecticide use [11].

One such species complex, whose biology is sporadically understood across broad swaths of the United States, is the billbug complex, a group of grass feeding weevils (Coleoptera: Curculionidae: *Sphenophorus* spp. Schönherr) [12]. Sixty-four described billbug species are native to North America, and although adults can be identified based on morphological characteristics [13], the larvae are morphologically indistinguishable. Our inability to accurately identify the cryptic, soil-dwelling larval stage of these weevils has limited our capacity to characterize the seasonal biology of the eleven turfgrass feeding species [12]. Because the seasonal biology of most species is poorly understood, our ability to effectively manage these pests on golf courses, athletic fields, home lawns and other turfgrass ecosystems, has been problematic. Billbug larvae feed within the stems, roots and crowns of turfgrasses causing desiccation and plant death, markedly reducing the aesthetic and functional quality of managed turf [14]. Currently, growers are faced with the challenge of managing these pests without a sound understanding of regional variation in pest species composition and seasonal biology. 

Efficient billbug management relies heavily on proper timing of synthetic insecticides or biological control approaches that often vary in their efficacy against specific life stages [15,16]. The number and timing of synthetic insecticide applications necessary for satisfactory control may also vary depending on the species present and the seasonal biology of each within a particular region. Unfortunately, the seasonal biology of most turfgrass-associated billbugs species has not been studied in detail, and the seasonal biology of the most well-studied species appears to vary geographically. Because of the sympatric distribution of several billbug species, our patchy understanding of their seasonal biology can make satisfactory management difficult to achieve in many regions. In Indiana there are four billbug species that are commonly associated with turfgrass (*S. parvulus* Gyllenhal, *S. venatus* Say, *S. minimus* Hart, and *S. inaequalis* Say), with the two most common species (*S. parvulus* and *S. venatus*) having very different life history strategies. While *S. parvulus* overwinters in Indiana in the adult stage, *S. venatus* overwinters in both the adult and larval stage [17]. In Utah, three species routinely infest turfgrass (*S. parvulus*, *S. venatus* and *S. cicatristriatus* Fåhraeus) and although the adults of each species are active from February to October, their larval phenology has not been documented in this region [18]. Further, the most widely studied species, *S. venatus*, displays a highly flexible seasonal biology that appears to vary regionally in the number of generations produced each year (1–6) and potentially the life stage structure or demographics of the overwintering cohort [14,17,19,20,21]. As such, holistic, regionally appropriate management strategies have been difficult to articulate. The development of a reliable approach for distinguishing billbug larval species identity would assist regional efforts to characterize billbug seasonal biology and support grower’s efforts to develop more prescriptive management programs. 

One potential avenue for reliably identifying billbug larvae is DNA barcoding, which involves the use of specific genes or genomic regions for species identification. DNA barcoding has shown good outcomes in differentiating between cryptic species and between species with cryptic developmental stages [4,11,22,23]. In particular, the cytochrome oxidase subunit I (COI) gene is commonly targeted for animals, and COI has also been widely applied in insect identification studies [24] that range from biodiversity [25] and food safety [26], to community ecology [27]. COI typically exhibits limited intraspecific variation, allowing researchers to reliably group members of the same species together, but demonstrates enough interspecific variation to separate different species [28]. It has been used previously to improve understanding of pest species complexes and differentiate even closely related species [24]. DNA barcoding has been widely used to support the development of more efficient and sustainable insect management programs, and there are at least two previous examples specific to turfgrass. By using DNA barcoding techniques to identify turfgrass-infesting *Phyllophaga* spp larvae, Doskocil et al. [11] laid the groundwork for later efforts aimed at characterizing the temporal and spatial distribution of these insects in Oklahoma [29]. Similar techniques were used by Duffy et al. [17], to identify the larval stage of several billbug species and clarify their seasonal phenology in Indiana. Biological information emerging from such studies may translate directly into extension programming that supports growers’ ability to develop and implement regionally tailored integrated pest management (IPM) strategies.

The findings of Duffy et al. [17] demonstrated that DNA barcoding can provide critical biological insights in support of billbug IPM goals on a local scale. In their study, Duffy et al. [17] included four species present in Indiana, *S. venatus*, *S. parvulus*, *S. inaequalis*, and *S. minimus,* and amplified three barcoding genes: COI (mitochondrial), 18S (nuclear ribosomal), and ITS2 (nuclear). Such multi-gene approaches have been previously developed [30], and using this approach, the researchers were able to detect that *S. venatus* overwinters in both the adult and larval stage, resulting in two distinct cohorts capable of damaging turfgrass during different times of the growing season. Although Duffy et al. [17] assessed the seasonal biology of the billbug species complex within one geographically defined area (Indiana), our ability to apply this technique across a larger geographic area, where intraspecific genetic variation may be much higher, remains unclear.

Since geographically driven genetic variation could influence the utility of DNA barcoding genes for resolving species identity [31], we assessed the utility of a DNA barcoding approach using three different genes (i.e., COI, ITS2, 18S). First, we hypothesized that by using a combination of three genetic loci we could characterize the intraspecific variation and interspecific divergence of billbug species across several states located in different regions of the U.S. (Indiana, Missouri, Utah, and Arizona). Secondly, as proof of concept, we hypothesized that by employing intensive billbug larval sampling and a simple morphometric measure (head capsule diameter) in conjunction with DNA barcoding, we could characterize the species composition and seasonal biology of the billbug complex across geographically disparate U.S. states. These aims will provide insights into billbug biology that could be used to develop regionally relevant, prescriptive monitoring and management programs for billbug pests in turfgrass systems.

## 2. Materials and Methods

### 2.1. Adult DNA Barcoding Reference Database

The first step towards assessing the utility of a DNA barcoding approach to differentiate cryptic billbug larval species was to create a reference database of adult billbug barcoding sequences. This reference database was used to characterize intraspecific variation and interspecific divergence of billbug species, and for comparison with larval sequences for species identification.

In order to include representation of the most common billbug species from different regions, billbug adults were collected during the growing season by hand, or using pitfall traps in Indiana (2016), Utah (2018), Missouri (2018), and Arizona (2018). Adult specimens were identified to species level based on morphological characters described in Johnson-Cicalese et al. [13], placed into glass vials containing 90% ethanol and stored at −20 °C until further processing. Each specimen was assigned a number and the corresponding species identity, collection location, and collection date were entered into the database.

The thorax and abdomen of each adult specimen were homogenized using a pestle, and DNA was extracted using the Qiagen^®^ DNAeasy Blood and Tissue kit following the standard protocol established by the manufacturer and specifically for animal tissue using spin-columns. We optimized this standard protocol for use with adult and larval billbug specimens using a 3 h initial incubation step at 56 °C following Duffy et al. [17]. DNA quality was assessed by visualizing genomic DNA on a 1% agarose gel. DNA concentration was measured using the Thermo Scientific™ NanoDrop™ one Microvolume UV-Vis Spectrophotometer. Samples with DNA concentrations above 50 ng/μL were diluted to reach concentrations between 20 ng/μL–50 ng/μL for optimal polymerase chain reaction (PCR). DNA was stored at −20 °C following extraction, dilution and PCR.

To assess the effectiveness of different genes to differentiate between billbug species, we amplified three commonly used barcoding genes: cytochrome oxidase subunit I (COI, mtDNA), internal transcribed spacer region 2 (ITS2, nrDNA), and 18S rRNA V3 region (nrDNA). Primer sequences and PCR conditions were established following the protocols of Duffy et al. [17]. Gel electrophoresis at 1% agarose in 1X TAE buffer was used to confirm amplification of PCR products from adult billbugs. The expected length of PCR products for each gene were 750 bp-COI, 650 bp-18S, and 250 bp-ITS2 based-on data from Duffy et al. [17]. Amplified products were then cleaned using the Exo SAP-IT PCR Product Cleanup Reagent™ following manufacturer protocols. After cleanup, samples were sent for Sanger Sequencing to the Purdue Genomics Core facility or to Genewiz (South Plainfield, NJ, USA).

Resulting forward and reverse sequences for each of the three barcoding genes were processed using the Aliview [32], alignment and editing software. The quality of nucleotide sequence was determined by examining individual chromatograms using the 4 peaks software [33]. To create a consensus sequence or contig, reverse sequences were reverse complemented and then aligned with the forward sequence. Primer sequences and low-quality base pairs were trimmed from the ends of the aligned sequences and then forward and reverse alignment were merged creating the consensus contig. The resulting length of consensus sequences used for further analysis were 640 bp (COI), 350 bp (18S), and 202–520 bp (ITS2). All sequences obtained from adult billbugs in this study were deposited in NCBI (GenBank accession numbers OK236222–OK236254, OK244534–OK244555, and OK244699–OK244728).

Consensus sequence alignment was performed using the MUSCLE algorithm [34] and included existing billbug sequences from Duffy et al. [17]. A phylogenetic analysis for non-coding proteins was carried out using MEGA software [35]. The evolutionary history was inferred by using the Maximum Likelihood method and General Time Reversible model in MEGA software with nodal branch support of 1000 bootstrap replicates. Initial tree(s) were obtained automatically by applying Neighbor-Joining and BioNJ algorithms to a pairwise distance matrix estimated using the Maximum Composite Likelihood (MCL) approach. Log Likelihood values were then used to select the trees with superior topology. A discrete Gamma distribution was used to model evolutionary rate differences among sites (each base pair). These initial trees were then edited using Mesquite software [36] and Inkscape [37] to improve their aesthetic quality for publication. A final phylogenetic tree following this pipeline was produced for each of the three sequenced genes separately (COI, 18S, and ITS2).

Due to the potential intraspecific variation in the DNA sequences of the species widely distributed through North America (*S. parvulus* and *S. venatus*), we assessed whether this variation could result in discrepancies in identification. A distance matrix of percent differences was assembled and translated into percent sequence similarity to compare sequences within *S. parvulus* and *S. venatus* to every other species. Graphs depicting sequence variation were constructed using the ggplot2 [38] package in R. In addition, we performed standard barcode gap analysis based on genetic distances calculated using the Kimura-2-parameter (K2P) model with the BarcordingR [39] package in R. Distributions of intra- and interspecific distances were visualized in a historgram plot, where the non-overlapping of interspecific and intraspecific genetic distance distribution is considered the barcoding gap that indicates clear species boundaries.

### 2.2. Larval Species Identification

To test the effectiveness of DNA barcoding for larval species identification and elucidate billbug seasonal biology, billbug larval sampling was performed throughout a portion of the growing season in Utah (21 May 2018 to 26 July 2018) and Indiana (3 June to 5 August 2020). These sampling dates were chosen based on the first appearance of larvae in the soil and proceeded over a 8–9 week period corresponding to peak larval activity. Soil cores were collected using a standard golf course cup-cutter (10.8 cm diameter) to a depth of 10 cm, and cores were carefully broken apart while searching for larvae. Collection locations are detailed in Table 1. Larvae were placed in 90–95% ethanol and stored at −20 °C for further processing. Each larval specimen was numbered, and the corresponding collection date and location were entered into a larval database.

To track larval development across the growing season, larval head capsule width was measured and recorded for all larvae and entered into the corresponding database. Larvae were dorsally imaged using a Leica DFC450 camera mounted onto a MC165C stereomicroscope and head capsule widths were measured using the Leica Application Suite version 4.2.0 (Leica Microsystems). After head capsule measurements were taken, DNA barcoding of the larval specimens was performed following the same protocol used for adults, including DNA extraction, PCR amplification, Sanger sequencing and processing of consensus sequences for each individual. Only the COI gene was sequenced for larval specimens since it provided the optimal combination of species resolution and sequencing success for adults.

We attempted to identify morphologically cryptic billbug larvae to species level using two methods based on DNA barcoding with the COI gene. First, we built a phylogenetic tree that included all larval and adult sequences following the same maximum likelihood approach previously described (see DNA Sequence Analysis section above). Phylogenetic tree-based identification was employed by observing where the larvae were located within well supported clades (bootstrap value > 70%) that included adult specimens. The second approach we used to identify larval specimens and assess intra- and inter-specific variation involved measuring average percent sequence similarity, which was carried out by comparing each larval sequence to all other larval and adult COI sequences. The length of the gene region used for sequence comparison for COI was 640 bp. Graphs depicting sequence variation were constructed using the R package ggplot2 and species identification was assessed by looking at the position of each data point within the graph. A threshold of 91.25% sequence similarity was used for larval species differentiation. This percentage was chosen based on observed COI sequence variation for *S. parvulus* larvae, the species showing the highest level of intraspecific variation.

### 2.3. Larval Seasonal Phenology Charts

As proof of concept, we developed phenology charts based on a combination of morphometric measures (larval head capsule width) in conjunction with larval DNA barcoding to characterize the seasonal biology of the billbug species complex from Indiana and Utah across a portion of the growing season. Larval phenology charts were created by including the head capsule width data, collection location, date of collection, and larval species identification based on our DNA barcoding methods using the COI gene. Two phenology charts were created, one each from Utah (2018) and Indiana (2020). The analytics system Statistica^®^ 13.3.0 [40] was used to develop the charts by plotting head capsule diameter (Y-axis) against day of year (X-axis) for each species present at a given location. Head capsule width measurements were used as a proxy for larval development, a method commonly used in Entomology [41]. Although billbug larval development proceeds through the course of five instars [18], we adopted the approach of previous studies [17,19] and binned billbug larvae as small (head capsule width < 1.0 mm), medium (1.0–1.7 mm), or large (above 1.7 mm).

## 3. Results

### 3.1. Adult DNA Barcoding Reference Database

As the first step towards developing a DNA barcoding method to identify morphologically indistinguishable billbug larva, we created a reference database of morphologically identified adult billbug species sequences. Then, we assessed the utility of three potential barcoding genes: COI, ITS2 and 18S. The adult DNA reference database included sequences from a wide variety of turfgrass feeding billbug species collected from different geographic locations. A total of ninety-seven adult sequences were obtained across all three potential barcoding genes (Table 1) and thirteen sequences were retrieved from the database created by Duffy et al. [17] (Table 1). COI produced the highest percentage of success in obtaining high-quality sequences across specimens (Table 1).

The phylogenetic tree constructed using adult COI sequences shows strong support for monophyly of every billbug species except *S. parvulus*, but within *S. parvulus* there was strong support for two subclades with bootstraps = 100% for both (Figure 1). ITS2 provided lower single gene resolution than COI (Figure 2) with strong support for *S. inaequalis* and *S. cicatristriatus* monophyletic clades (bootstrap values ≥ 95%) but weak support for *S. parvulus* and *S. venatus*, despite that they were grouped into monophyletic clades (bootstrap values ≥ 49%). ITS2 failed to group *S. minimus* into a monophyletic clade. This gene also showed inconsistent amplification compared to COI and produced only twenty-two usable sequences (Table 2). Finally, 18S generated almost no resolution with unreliable species identification, and failed to group species into monophyletic clades (Figure 3; bootstrap values ≥ 0%).

Our results indicate the COI barcoding region reliably differentiated between species regardless of the geographic location from which they were collected (Figure 1). Each species (*S. parvulus* and *S. venatus*) formed monophyletic clades that were not separated by location (bootstrap values of 66 and 100% for *S. parvulus* and *S. venatus*, respectively) (Figure 1). However, *S. parvulus* did separate into two well-supported sub-clades with bootstrap values of 100% each. These two clades did apear to have some underlying geographic structure with one clade composed entirely of midwestern speimens (100% Indiana and Missouri) whereas the other was compost almost entirely of specimens form the intermountain west (91% Utah, 9% Indiana). When comparing the average percent sequence similarity of each adult specimen with all others within the same species group, *S. parvulus* contained more intraspecific sequence variation (92.5–99% sequence similarity) than *S. venatus* (95–99% sequence similarity) (Figure 4). Wether or not obseved variation in sequence similarity among *S. parvulus* specimens reflects the existence of a criptic species or the manifestation of geographic/allopatric forces remains to be investigated. For the purposes of this study, intraspecific variation resulting in sequence similarity above 90% (Figure 4), did not interfere with the ability of COI to differentiate currently recognized species using our DNA barcoding method. Additional barcode gap analysis based on K2P genetic distances further supports the ability of COI to reliably differentiate between the billbug species used in this study (see Appendix A
Figure A1). Due to the usefulness of 90% raw sequence similarity values for differentiating adult specimens, this percentage was also used as a threshold for larval identification in later analyses.

### 3.2. Larval Species Identification

A total of 138 billbug larval specimens were collected for comparison against the billbug adult DNA reference database of COI sequences (Table 3). We were able to identify morphologically cryptic billbug larvae collected from Indiana and Utah to species level (Figure 5 and Figure 6; Table 2). Using a phylogenetic approach, larval COI sequences formed monophyletic clades that aligned with the adult billbug sequences (bootstrap values above 99%) (Figure 5). In addition, average percent sequence similarity confirmed that our DNA barcoding approach could reliably identify billbug larvae given existing intraspecific variation (>91.25–99% average sequence similarity) (Figure 6). Intraspecific variation in larval COI sequences ranged across species; *S. parvulus* (>91.25 average sequence similarity), *S. venatus* (>97.5 average sequence similarity) and *S. minimus* (>99% average sequence similarity) (Figure 6), but could be partially driven by differences in the number of specimens of each billbug species available for analysis.

### 3.3. Larval Seasonal Phenology Charts

As proof of concept that elucidating the seasonal biology of billbugs could be achieved by combining our DNA barcoding approach with head capsule width data, we developed larval seasonal phenology chars for Indiana and Utah This technique has previously been reported by other authors attempting to characterize billbug larval development [17]. The resulting charts allowed us to visualize differences in species composition whereas head capsule diameters provide a visual representation of how larval development of different billbug species proceeded across a portion of the growing season in these two different regions of the U.S. (Figure 7).

Larval specimens collected from Utah during the summer of 2018 consisted of, *S. venatus* and *S. parvulus*, whereas larval specimens collected from Indiana included *S. venatus*, *S. parvulus*, and *S. minimus* (Table 2). All larval specimens from Utah were collected from stands of Kentucky bluegrass (cool-season grass) while in Indiana they were collected from warm- and cool-season grasses. In Indiana, 68% of the specimens sequenced in cool-season grasses were identified as *S. parvulus*, 12% as *S. venatus*, and 20% as *S. minimus* (Table 3). In warm-season grasses 92% were identified as *S. parvulus*, 6% as *S. venatus*, and 2% as *S. minimus* (Table 3).

## 4. Discussion

Turfgrass infesting billbugs represent an economically significant species complex consisting of no fewer than eleven North American species [18,42,43]. The composition of the billbug species complex varies regionally [13], with the geographic distribution of several species overlapping over large portions of their range. Although the seasonal biology of many species has not been studied in detail, the seasonal biology of the most well-studied species also appears to vary geographically. Additionally, it is currently impossible to accurately identify the larval stage to species level based on morphological characters. As a result, efforts to disentangle the seasonal biology and life history of these insects have been impossible to achieve in areas where mixed-species populations are common [17]. Our study aimed to provide a molecular tool that will allow researchers across the U.S. to accurately identify the cryptic, soil-dwelling, larval stage and apply that tool to better understand billbug seasonal biology—a prerequisite for developing sound monitoring and management strategies.

### 4.1. Development of a Reliable DNA Barcoding Tool for Sphenophorus in the U.S.

In order to precisely identify unknown specimens, a robust reference database is essential in DNA barcoding [44]. It requires expanding sample sizes beyond what is available in public databases such as GenBank and the Barcode of Life Data Systems (BOLD) to avoid ambiguous results [45], and has proven essential for advancements in metabarcoding approaches that employ next-generation sequencing technologies. Previous work by Duffy et al. [17] demonstrated that a combination of three different barcoding genes (COI, ITS2, and 18S) could be used collectively to identify the morphologically indistinguishable larvae of four different billbug species in Tippecanoe County, IN. However, we suspected that our goal of producing a robust, well-supported and broadly applicable phylogenetic identification tool would require a broader, more geographically diverse sampling of *Sphenophorus* taxa. For this reason, our reference database of adult billbug sequences included six species, four of which are regionally dominant in the U.S.: *S. parvulus* (North), *S. venatus* (Southeast), *S. phoeniciensis* (Southwest), and *S. cicatristriatus* (Rocky Mountain) [43]. Sequenced specimens also originated from four, geographically diverse states (Utah, Indiana, Arizona, and Missouri), with the two most widely distributed species (*S. venatus* and *S. parvulus*) providing sequences from three of these states (Utah, Missouri, and Indiana).

Based on earlier reports [17], we hypothesized that a combination of three barcoding genes (COI, 18S, ITS2) may be required to accurately characterize intraspecific variation and interspecific divergence of billbug species across the U.S. regions included in the current study. The limited dispersal capabilities of billbug adults [18] suggests that Midwestern billbugs could be genetically dissimilar to those inhabiting other parts of the U.S., potentially creating a variable geographic signature that could limit the utility of any single barcoding gene. However, contrary to our prediction, COI alone provided the highest, single gene resolution, and was able to consistently separate all six billbug species into monophyletic clades, regardless of geographic variation. These findings differ from those of other researchers who have reported a lack of success in using COI as a single barcoding gene in other insect groups and postulated the need to include additional genes [31,44,46]. The use of a single gene in DNA barcoding studies has been contested in the past due to known limitations that include multiple mitochondrial gene haplotypes (heteroplasmy) and nuclear pseudogenes of the mitochondria genome (NUMT) [24]. However, Rubinoff et al. [31] demonstrated that as long as COI is being used in a well-studied group of insects with known characteristics for adult species differentiation, this gene alone may provide the resolution required to effectively distinguishes species. The current study complies with the general guidelines set forth by Rubinoff et al. [31] and supports the idea that COI alone may provide the resolution necessary for species differentiation among *Sphenophorus* taxa across broad swaths of the continental U.S. Further, these results indicated that COI could be used to identify morphologically indistinguishable billbug larvae, including widely distributed species, potentially strengthening the utility of this single gene approach for clarifying regional differences in seasonal biology.

Although results from the current study demonstrate that COI alone works well for billbug species identification, we also considered the utility of additional genes. In addition to COI, we individually assessed the utility of 18S and ITS2 as barcoding genes for *Sphenophorus*. Even though in some cases the use of 18S has been plagued with a low success rate in PCR amplification, it did provide resolution of scale insects when amplification was successful [47] and in ticks at the genera level [48]. Likewise, ITS2 has provided adequate resolution for species differentiation in other groups of insects such as *Anopheles* spp. [5], calliphorids [44], braconids [49], among others. However, ITS2 presents some documented difficulties, such as indels, that may affect alignment [50], and intragenomic variants capable of complicating the Sanger sequencing reaction [49].

In the current study, 18S did not provide the resolution necessary to differentiate billbug species due to low interspecific variation. Both 18S and ITS2 suffered from lower PCR success rates compared to COI. Similarly, ITS2 fell short of the single gene resolution provided by COI, grouping *S. parvulus* and *S. venatus* sequences into separate clades, but with lower bootstraps values and failing to group together members of *S. minimus*. Although problems articulated with the use of ITS2 could potentially be addressed through the use of next-generation sequencing techniques [49], difficulties in using 18S may be more challenging to manage. As such, neither 18S nor ITS2 alone appear to be viable candidates for the development of a robust, well-supported and broadly applicable phylogenetic identification tool for *Sphenophorus*.

### 4.2. DNA Barcoding Tool Provides Insight into Billbug Biology

Because of the resolution provided by COI, we used this barcoding gene in conjunction with billbug larval sampling and head capsule data to characterize a portion of the seasonal biology of the billbug complex in two different geographic regions of the U.S. Using this approach, we were able to visualize species composition and characterize larval development for several billbug species across a portion of the growing season in Utah and Indiana. The resulting seasonal phenology charts support the idea that the life history of turfgrass-inhabiting billbugs, including the cryptic, soil dwelling larvae, can be brought into focus across different U.S. regions using this methodology. Although the data presented herein (May-August) represent only a portion of the entire growing season (March-November), we could clarify the species composition and variation in the development of each species’ destructive, soil-dwelling larval stage. This information is crucial for developing efficient pest management programs, as the effectiveness of different chemical and biological management approaches hinges on knowledge of the target stage.

As evidence for how this new approach may reveal important insights into billbug biology, we catalogued two somewhat unexpected findings. First, we were able to identify *S. parvulus* as the primary species infesting warm-season (C4) grasses at the Indiana location, despite the presence of *S. venatus* within the same turf stand. *S. parvulus*’ distribution is closely linked to areas where Kentucky bluegrass is grown which has resulted in the general working assumption that *S. parvulus* is primarily associated with cool-season grasses [43]. Second, our larval sampling efforts revealed *S. venatus* as a secondary pest species infesting cool-season grasses (Kentucky bluegrass) at both locations, despite its common association with, and documented damage in zoysiagrass and bermudagrass [43]. Moreover, *S. parvulus*, *S. venatus*, and *S. minimus* larvae were collected from the same stand of Kentucky bluegrass at the Indiana DTRC location. These findings support the utility of DNA barcoding as a larval identification tool and underscore that common billbug-host associations, or the mere presence of morphologically identifiable adults may not translate directly to soil-dwelling larval populations that are responsible for the majority of turfgrass damage.

Our finding supports the idea that billbug management should be anchored in biology and that species composition and seasonal biology investigations are essential for effective billbug management. Billbug control strategies rely heavily on the proper matching and timing of synthetic insecticides or biological control approaches targeting a particular billbug life stage, with active ingredients, application timing and number of applications required to provide satisfactory control varying depending on the seasonal biology of the target species. In the Midwest, the application of DNA barcoding revealed that the seasonal biology and population dynamics of the two most common billbug species differ in ways that required fundamentally different approaches toward monitoring and management [17]. In regions where billbug species composition and seasonal biology is still unknown, our DNA barcoding tool will be useful for disentangling species identity and clarifying seasonal population dynamics thereby supporting ongoing efforts to develop efficient management strategies.

## 5. Conclusions

The current study advances our ability to accurately identify the destructive, soil-dwelling larval stage of *Sphenophorus* taxa, even in cases where geographically driven genetic variation may be expected. With a more robust DNA based larval identification tool in place, this research may be leveraged to close important gaps in our understanding of billbug seasonal biology and species composition throughout the continental U.S. However, additional sampling across the US is needed to further explore geographic patterns of variation and provide a more in depth analysis of billbug evolutionary history and taxonomy. Futhermore, since efforts to create effective and efficient management strategies are undermined by making “common sense” associations between presence of particular adult species and their favored host plants, our findings emphasize the importance of identifying the damaging larval stage. The COI gene alone was able to differentiate between billbug species regardless of where they were collected, and we were able to confidently identify billbug larvae using this single mitochondrial gene. By combining larval identification, collection dates, and morphometric data (head capsule diameter), the regionally variable life history of turfgrass-inhabiting billbugs can be clarified and used to anchor management programs. Future efforts are still needed to test the robustness of COI across additional species and regions that were not included in the current study.

## Figures and Tables

**Figure 1 insects-12-00930-f001:**
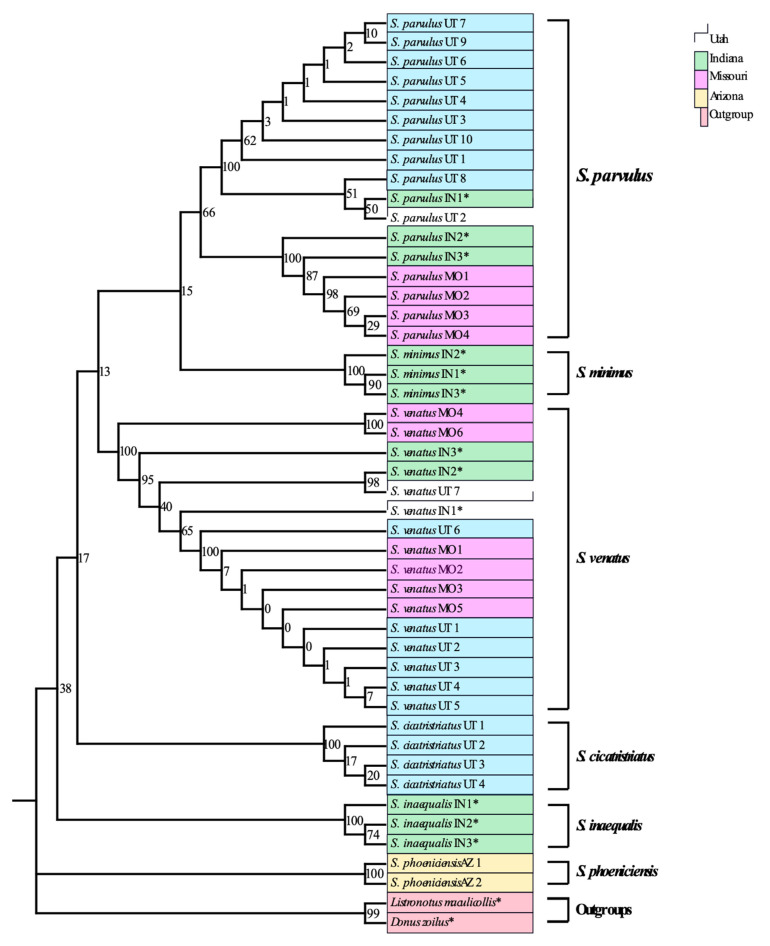
Maximum likelihood tree of COI sequences from *Sphenophorus parvulus, S. venatus*, *S. minimus*, *S. inaequalis*, *S. cicatristriatus*, and *S. phoeniciencis* adults. Collection location is represented by color blocks; Utah (blue), Indiana (green), Missouri (purple), Arizona (yellow), outgroups (red). Replicate numbers are indicated to the right of the scientific name and collection state (Utah = UT, Indiana = IN, Missouri = MO, Arizona-AZ). Numbers at nodes are bootstraps values (1000 bootstrap replicates as percentages). * indicates sequences obtained from [17].

**Figure 2 insects-12-00930-f002:**
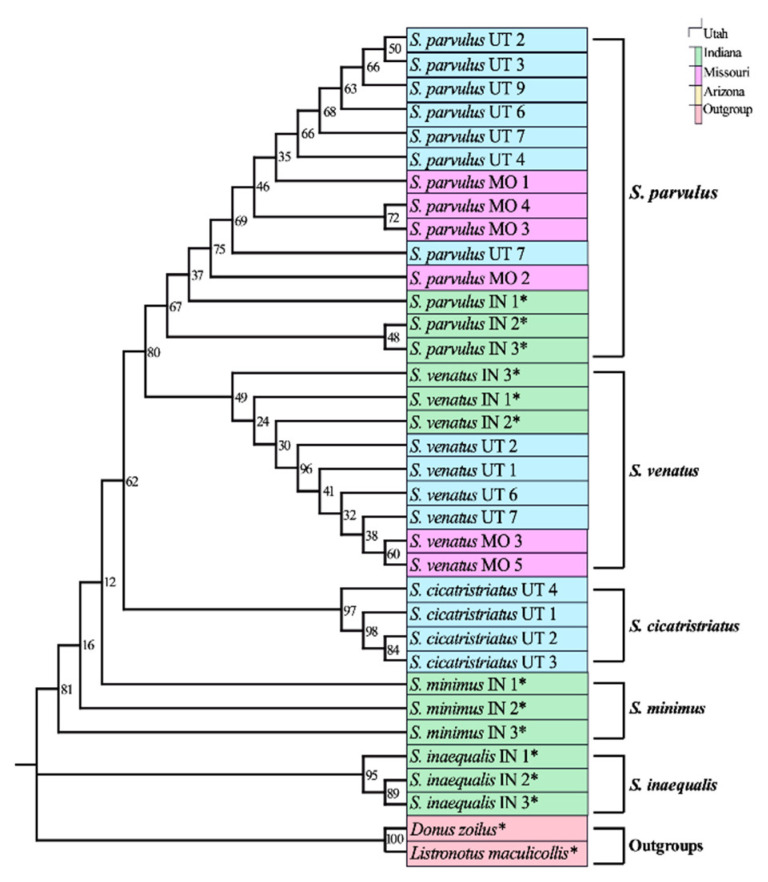
Maximum likelihood tree of ITS2 sequences from *Sphenophorus parvulus, S. venatus*, *S. minimus*, *S. inaequalis*, and *S. cicatristriatus*. Collection location is represented by color blocks; Utah (blue), Indiana (green), Missouri (purple), Arizona (yellow), outgroups (red). Replicate numbers are indicated to the right of the scientific name and collection state (Utah = UT, Indiana = IN, Missouri = MO, Arizona-AZ). Numbers at nodes are bootstraps values (1000 bootstrap replicates as percentages). * indicates sequences obtained from [17].

**Figure 3 insects-12-00930-f003:**
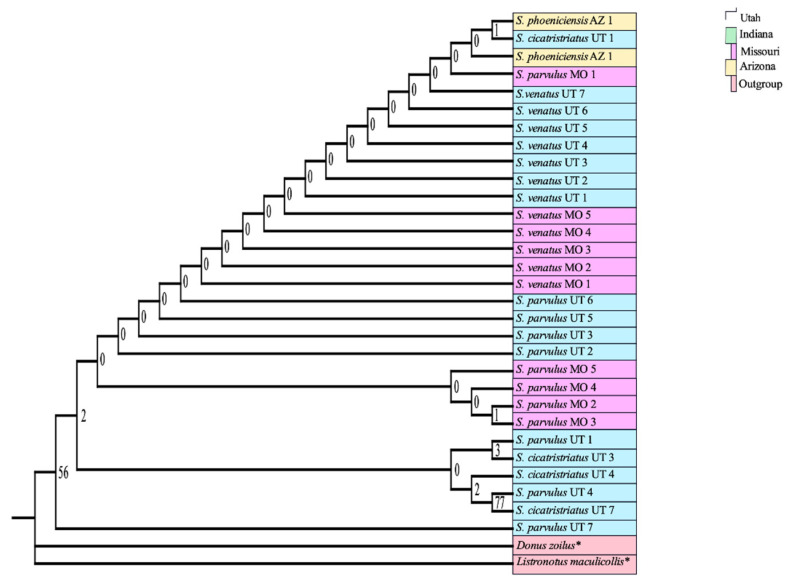
Maximum likelihood tree of 18S sequences from *Sphenophorus parvulus*, *S. venatus*, *S. inaequalis*, and *S. cicatristriatus*. Collection location is represented by color blocks; Utah (blue), Indiana (green), Missouri (purple), Arizona (yellow), outgroups (red). Replicate numbers are indicated to the right of the scientific name and collection state (Utah = UT, Indiana = IN, Missouri = MO, Arizona-AZ). Numbers at nodes are bootstraps values (1000 bootstrap replicates as percentages). * indicates sequences obtained from [17].

**Figure 4 insects-12-00930-f004:**
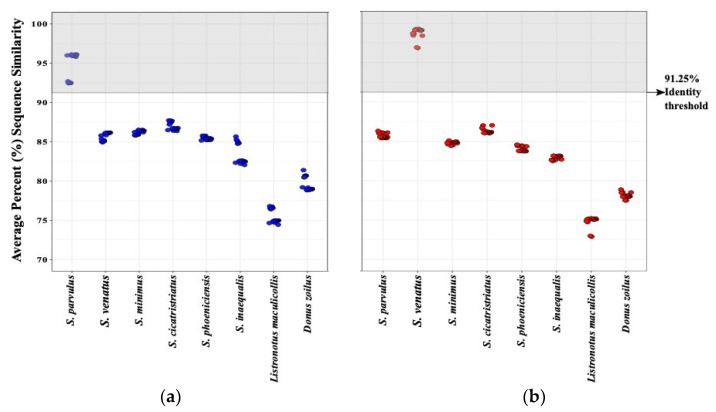
Pairwise distances between adults measured as average percent sequence similarity of the COI gene (640 bp) show successful species identification above a 91.25% similarity threshold—a visual representation of the method for comparing unknown samples to a reference DNA barcode database for species identification. Graph (**a**) depicts *Sphenophorus parvulus* specimens collected from several locations (blue dots) and graph (**b**), *S. venatus* specimens collected in several locations (red dots). To calculate average percent sequences similarity, each individual sequence from *S. parvulus* (**a**) or S. venatus (**b**) was compared to all other sequences in our reference database: *S. parvulus*, *S. venatus*, *S. minimus*, *S. cicatristriatus*, *S. phoeniciensis*, *S. inaequalis*, *Listronotus maculicollis* and *Donus zoilus*. The area in gray highlights the range of sequence similarity above the 91.25% threshold used for species identification. Data points falling above the threshold value (in grey area) are assigned to the species shown below on the x-axis.

**Figure 5 insects-12-00930-f005:**
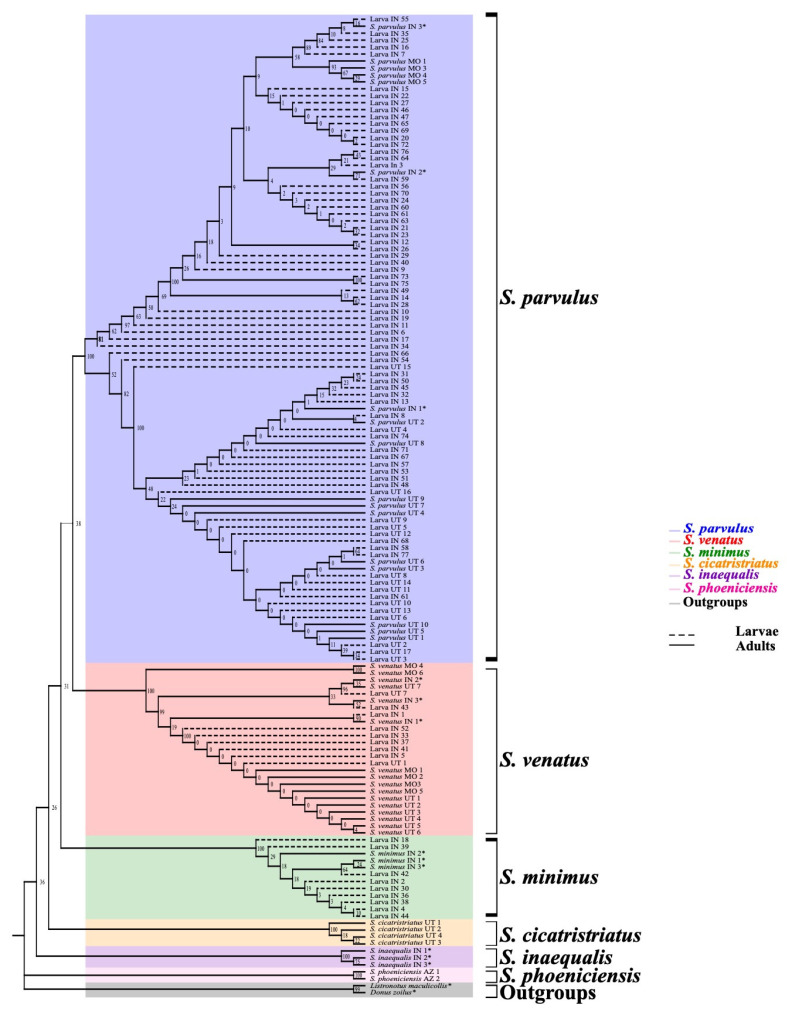
Maximum likelihood tree of COI sequences from adults and larvae. Larvae are represented as dashed lines and were identified based on their position in the tree with bootstraps values ≥50%. Larvae were successfully grouped with adult species included in the reference database, species groupings are represented by color blocks: *Sphenophorus parvulus* (blue), *S. venatus* (red), *S. minimus* (green), *S. inaequalis* (purple), *S. cicatristriatus* (orange), and *S. phoeniciencis* (pink). Replicate numbers are indicated to the right of the scientific name and collection state (Utah = UT, Indiana = IN, Missouri = MO, Arizona-AZ). Numbers at nodes are bootstraps values (1000 bootstrap replicates as percentages). * indicates sequences obtained from [17].

**Figure 6 insects-12-00930-f006:**
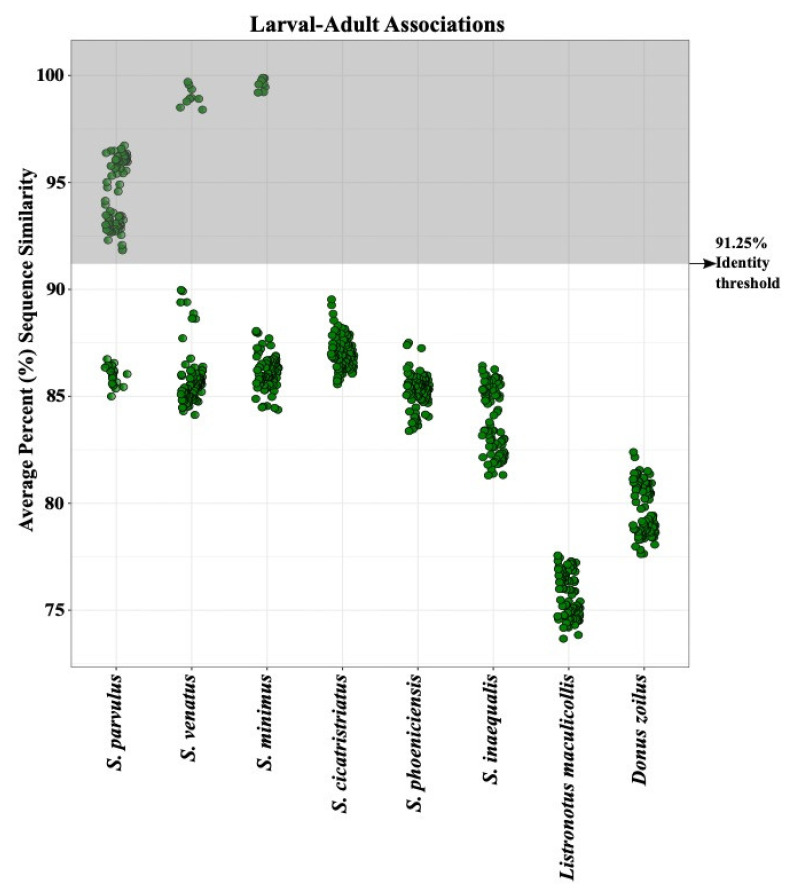
Pairwise distances between larvae and adults (Utah 2018 and Indiana 2020) measured as average percent sequence similarity of the COI gene (640 bp). Each green dot represents a larval sequence matched against the adult reference database sequences (i.e., above the 91.25% mark in the gray area of figure). Larval sequences are compared to all sequences of each species included in the reference database: *Sphenophorus parvulus*, *S. venatus*, *S. minimus*, *S. cicatristriatus*, *S. phoeniciensis*, *S. inaequalis*, *Listronotus maculicollis*, *Donus zoilus*.

**Figure 7 insects-12-00930-f007:**
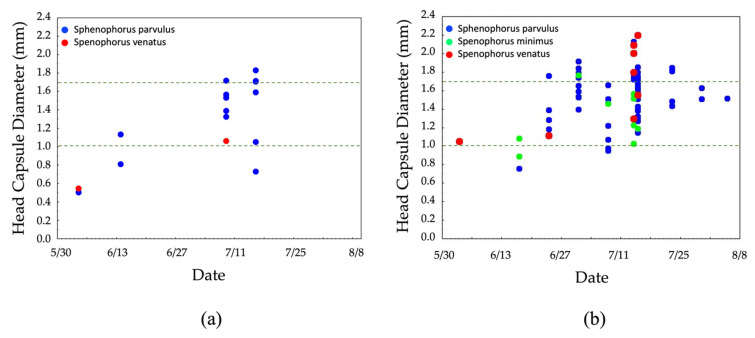
Seasonal phenology charts of billbug larvae collected from (**a**) Utah 2018 and (**b**) Indiana 2020. We adopted the approach of previous studies [17,19] binned larvae as small (head capsule diameter < 1.0 mm of), medium (1.0–1.7 mm), or large (above 1.7 mm).

**Table 1 insects-12-00930-t001:** Predominant turfgrass cover and soil types associated with billbug larval collection locations in Utah and Indiana.

U. S. State	Site (Coordinates)	Abbreviated Name	Turfgrass Cover	Turfgrass Cultivar	Soil Type
Utah	Logan Country Club	LCC	Kentucky bluegrass	Unknown	Silty Loam
	(41.7445° N, 111.7949° W)				
	Greenville Research Farm	GRF	Kentucky bluegrass	Unknown	Silty Loam
	(41.7664° N, 111.8105° W)				
	Utah State University Greenhouse	USUG	Kentucky bluegrass	Unknown	Silty Loam
	(41.7571° N, 111.8133° W)				
Indiana	Bimel Practice Center	BPC	Bermudagrass	Patriot	Silty Clay Loam
	(40.4376° N, 86.9178° W)				
	Daniel Turfgrass Research Center	DTC	Kentucky bluegrass	Park	Silty Clay Loam
	(40.4411° N, 86.9317° W)				
	Daniel Turfgrass Research Center	DTC	Zoysiagrass	Meyer	Silty Clay Loam
	(40.4415° N, 86.9325° W)				

**Table 2 insects-12-00930-t002:** Summary of DNA barcoding results used to create a reference sequence database from adult billbug (*Sphenophorus* spp.) species collected in Indiana, Utah, Missouri, Arizona, and two outgroup species. The total number of DNA extractions performed, successful sequencing of PCR products (% Success) and total number of sequences included in the reference database are shown for each barcoding gene (COI, 18S and ITS2).

Species	DNA Extractions	COI	% Success	18S	% Success	ITS2	% Success
*S. parvulus*							
Utah	14	10	71.4	7	50.0	7	50.0
Missouri	12	5	41.6	5	41.6	4	33.3
Indiana	0	3 *		0		3 *	
*S. venatus*							
Utah	15	12	80.0	7	46.6	4	26.6
Missouri	16	6	37.5	5	31.2	2	12.5
Indiana	0	3 *		0		3 *	
*S. minimus*							
Indiana	0	3 *		0		3 *	
*S. inaequalis*							
Indiana	0	3 *		0		3 *	
*S. cicatristriatus*							
Utah	17	10	58.8	4	23.5	4	23.5
*S. phoeniciensis*							
Arizona	2	2	100.0	2	100.0	1	50.0
*Listronotus maculicollis* *	0	1 *		1 *		1 *	
*Donus zoilus* *	0	1 *		1 *		1 *	
Total sequences		57		30		23	
Avg %			64.8		48.82		32.6

* Sequences obtained from previous study by [17].

**Table 3 insects-12-00930-t003:** Summary of larval specimens collected at Utah 2018 and Indiana 2020 in cool- and warm-season turfgrass. The total number of each species identified at each location and type of grass (* cool-season or ° warm-season) is included.

					**Total of Larvae Identified Per Species**		
Location	Year	Grass Type	Total of Larvae Collected	Total of Larvae Sequenced	*S. parvulus*	*S. venatus*	*S. minimus*
Utah	2018						
		Cool	24	17	15	2	0
		Warm	0	0	0	0	0
Indiana	2020						
		Cool	63	41	28	5	8
		Warm	51	36	33	3 ^*^	
Total			138	94	76		

* Cool-season = Kentucky bluegrass *Poa pratensis.* Warm Season = Bermudagrass *Cynodon dactylon* ‘Patriot’ and Zoysiagrass *Zoysia japonica* ‘Meyer’.

## Data Availability

The data presented in this study are openly available in Genbank and can be accessed through the following accession numbers OK236222–OK236254, OK244534–OK244555, and OK244699–OK244728.

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
