# Peer review of "Characterizing Billbug (*Sphenophorus* spp.) Seasonal Biology Using DNA Barcodes and a Simple Morphometric Analysis"

_insects, 2021, doi:10.3390/insects12100930_

Round 1

Reviewer 1 Report

Dear authors

Thank you for this very interesting study! I have only some minor comments mostly regarding the sampling and the presentation of your results.

Author Response

We have addressed the reviewer’s comments, which can be found incorporated in the tracked changes revised version of the manuscript. Specifically, we incorporated changes throughout the manuscript that improved clarity by including additional details to methods and figures (e.g., details added on lines 145-151 & 205-207; new Table 1 with sampling information; Figs 4&6 revised legends, Fig 5 updated), additional references were suggested (e.g., lines 61,63, 149, 395) and additional description of results addressing patterns of observed sequence variation (lines 302-320).

Reviewer 2 Report

I read carefully the submitted article  titled " Characterizing billbug (Sphanophorous spp.) seasonal biology using DNA barcodes and a simple morphometric analysis" that is a nice contribution  to the identification  of larval stages of billbug specimens collected in several States of North America (USA).  The identification at species level was obtained by methods which combine molecular approach by using DNA barcodes technique and a morphometric system referring to larval head capsule diameter (width). The interesting results obtained  can provide reliable tools to better understand the biology and to draft IPM programs for these insects. All the sections  of the mns -introduction,m aterials and methods, results , discussion and conclusions- have been clearly recorded and exhaustively commented . I only noticed some minor errors which have been pointed out in the review.I ask the AA. to consider them before re-submitting the mns. Please, look at the attached file.

Author Response

We have addressed the reviewer’s comments, which can be found incorporated in the tracked changes revised version of the manuscript. Specifically, we incorporated requested changes by including additional details to methods used for DNA extraction (lines 154-161), sampling location coordinates (see new Table 1) as well as additional references were suggested (e.g. line 102).

Reviewer 3 Report

The manuscript “Characterizing billbug (Sphenophorus spp.) seasonal biology using DNA barcodes and a simple morphometric analysis” (Insects-1365342) by Rodriguez-Soto and co-authors analyze the usefulness of three molecular markers (partial COI = DNA barcode sensu Hebert, partial ITS2, partial 18S) to discriminate and identify a number of billbug species (Sphenophorus spp.) in the United States. In addition, larval head capsule measurements were used to characterize regional variation in billbug species and to develop larval phenology maps.

In my eyes, the topic of this manuscript is interesting and appropriate for “Insects”. The used molecular techniques are sound, the presented results are of relevance and highlight the proposed use of DNA barcoding as approach of choice in modern species identification. However, there are some parts that have to be modified, changed or added.

Major points

  • The combination of different sequence markers in terms of (cryptic) species identification is not new (e.g., Frontiers in Zoology 2010, 7: 26). I think this should be mentioned/discussed.
  • As a consequence of the concerted evolution of the tandem-arranged nuclear rRNA-coding genes, these genes are highly uniform, and there should be no (intraspecific) variation. Various studies have demonstrated that even single substitutions, which typically are found in hypervariable regions (= expansion segments), indicate closely related but distinct species. Therefore, it would be good to know which region of the 18S rRNA gene has been analysed.
  • I recommend that the authors should create a project on the BOLD workbench (boldsystems.org) to (re)analyze their CO1 data set, using the software tools offered there, e.g., the BIN approach (BOLD; Ratnasingham and Hebert (2013): PLOS ONE 8: e66213). The workbench is very useful and represents the state-of-the-art method to analyze DNA barcode data sets. Interestingly, a quick look at BOLD revealed a number of already published COI barcodes of Sphenophorus venatus, S. minimus and S. parvulus, including specimens from U.S. I think you should include these data in your analysis … Furthermore, I miss a presentation and discussion of intraspecific and interspecific distances based K2P values, as it has become a standard in the analysis of DNA barcodes.
  • Classical tree topologies are not appropriate to visualize intraspecific variabilities in relation to different geographic regions (if the distances are not extremely high; see phylogeographic studies). As consequence I strongly recommend to apply statistical parsimony networks. Such networks allow a better presentation of the distances between closely-related haplotypes from different regions than classical tree topologies. Popular programs as PopART (http://popart.otago.ac.nz) are easy to use. Such networks will enhance the presentation of the results significantly.
  • Comprehensive sequence libraries represent an essential element in uprising modern bioassessment studies based on high-throughput technologies (metabarcoding). A discussion of this future application of DNA barcoding is unfortunately missing but highly relevant.

Please see some specific comments made via sticky notes on the PDF file of the manuscript.

Author Response

Point 1: The manuscript “Characterizing billbug (Sphenophorus spp.) seasonal biology using DNA barcodes and a simple morphometric analysis” (Insects-1365342) by Rodriguez-Soto and co-authors analyze the usefulness of three molecular markers (partial COI = DNA barcode sensu Hebert, partial ITS2, partial 18S) to discriminate and identify a number of billbug species (Sphenophorus spp.) in the United States. In addition, larval head capsule measurements were used to characterize regional variation in billbug species and to develop larval phenology maps.

In my eyes, the topic of this manuscript is interesting and appropriate for “Insects”. The used molecular techniques are sound, the presented results are of relevance, and highlight the proposed use of DNA barcoding as an approach of choice in modern species identification. However, there are some parts that have to be modified, changed, or added.

Major points

  • The combination of different sequence markers in terms of (cryptic) species identification is not new (e.g., Frontiers in Zoology 2010, 7: 26). I think this should be mentioned/discussed.

Response 1: We have added the suggested reference and additional text addressing the use of different sequence markers to identifying cryptic species (see lines 118-121).

  • Point 2: As a consequence of the concerted evolution of the tandem-arranged nuclear rRNA-coding genes, these genes are highly uniform, and there should be no (intraspecific) variation. Various studies have demonstrated that even single substitutions, which typically are found in hypervariable regions (= expansion segments), indicate closely related but distinct species. Therefore, it would be good to know which region of the 18S rRNA gene has been analyzed.

Response 2: We added information in the methods indicating the primers used to target the V3 region of the 18s rRNA (line 164).

  • Point 3: I recommend that the authors should create a project on the BOLD workbench (boldsystems.org) to (re)analyze their CO1 data set, using the software tools offered there, e.g., the BIN approach (BOLD; Ratnasingham and Hebert (2013): PLOS ONE 8: e66213). The workbench is very useful and represents the state-of-the-art method to analyze DNA barcode data sets. Interestingly, a quick look at BOLD revealed a number of already published COI barcodes of Sphenophorus venatus, S. minimus and S. parvulus, including specimens from U.S. I think you should include these data in your analysis … Furthermore, I miss a presentation and discussion of intraspecific and interspecific distances based K2P values, as it has become a standard in the analysis of DNA barcodes.

Response 3: We agree with the reviewer that the BOLD database is a highly useful platform for analysis of barcode data sets. We followed the reviewer’s suggestion and examined the Sphenophorus COI sequences in the BOLD database for additional sequences from turfgrass associated billbug species collected in the US. We found only 5 relevant sequences from S. venatus collected in OK and MS. There was considerable redundancy between BOLD entries mined from NCBI sequences and those we have already included in our analysis from Duffy et al (our sequences). The majority of the 69 total Sphenophorus BOLD entries are from species not relevant to our study because they are not turfgrass-associated species and/or were not collected in the US. Therefore, although the reviewer makes a good suggestion, we feel re-analysis of our entire dataset is not warranted. Greater than 5 sequences from a broader range of locations in the US would be needed to perform a more robust analysis addressing geographic/regional differences.

Overall, the reviewer makes several excellent points regarding the need for additional sequence data to explore geographic patterns of variation and use of additional measures of sequence variation (e.g. K2P) for further phylogenetic analysis  – however, these are outside the scope of this manuscript, which focuses on using DNA barcoding to identify cryptic larval stages and begins to develop a better understanding of billbug seasonal biology relevant for improving management decisions. Therefore, we have added additional discussion addressing the reviewer’s comments [lines 507-510]. We now highlight the need for additional sampling across the US and further in-depth studies addressing billbug evolutionary history and taxonomy.

  • Point 4: Classical tree topologies are not appropriate to visualize intraspecific variabilities in relation to different geographic regions (if the distances are not extremely high; see phylogeographic studies). As consequence I strongly recommend to apply statistical parsimony networks. Such networks allow a better presentation of the distances between closely-related haplotypes from different regions than classical tree topologies. Popular programs as PopART (http://popart.otago.ac.nz) are easy to use. Such networks will enhance the presentation of the results significantly.

Response 4: Figures 4 & 6 show intraspecific variability (and the “barcoding gap”) that aligns with modern DNA barcoding methods using a threshold of sequence similarity for taxonomic identification within communities (e.g. microbiome metabarcoding studies). While we agree with the reviewer that haplotype networks are an effective way to visualize intraspecific variation and even potential geographic patterns, we feel these networks are redundant with current figures and would not add new information beyond what is already presented. We thank the reviewer for the suggestion, but including haplotype networks for the 2 species for which we have sufficient samples (S. parvalus, S. venatus) is not warranted.

  • Point 5: Comprehensive sequence libraries represent an essential element in uprising modern bioassessment studies based on high-throughput technologies (metabarcoding). A discussion of this future application of DNA barcoding is unfortunately missing but highly relevant.

Response 5: We agree with the reviewer that modern metabarcoding approaches that use NGS technology also must rely on robust sequence libraries. We have included this point in the discussion [line 403-404], however a broader discussion of this topic is beyond the scope of the current study.

Point 6: Please see some specific comments made via sticky notes on the PDF file of the manuscript.

Response 6:We have addressed the reviewer’s additional comments, which can be found in the tracked changes revised version of the manuscript. Specifically, we incorporated changes throughout the manuscript that improved clarity by including additional details and references where suggested.

Round 2

Reviewer 3 Report

It is nice to see that the authors accepted most comments. However, some important aspects are unfortunately still missing.

Point 3: In terms of the already published data at BOLD I agree that the addition of only five additional sequences/specimens is not useful. However, if you use DNA barcoding as molecular method specimen identification, you should do this in the broadly commonly accepted way: creation of a BOLD project, an analysis based on K2P parameter and NJ topology, barcoded gap analysis etc to provide a comparability to other studies. Molecular identification does not rely on methods that are used in molecular phylogenetics. As the authors point out, they focus on the identification of larvae: in doing so, they should perform approaches that focus on that. Unfortunately, this is not given yet.

Furthermore, keep in mind that even single substitutions within 18S sequences – and in particular within expansion segments – indicate closely related but distinct (!) species as consequence of the concerted evolution history of these genes.

Author Response

Manuscript ID: insects-1365342

Title: Characterizing billbug (Sphenophorus spp.) seasonal biology using DNA barcodes and a simple morphometric analysis

Authors: Marian M. Rodriguez-Soto, Douglas S. Richmond, Ricardo A. Ramirez, Xi Xiong, Laramy S. Enders

Responses to reviewer comments are highlighted in blue.

In addition, we provided a revised version of our manuscript with tracked changes.

Reviewer 3 Comments – Round 2:

Point 3: In terms of the already published data at BOLD I agree that the addition of only five additional sequences/specimens is not useful. However, if you use DNA barcoding as molecular method specimen identification, you should do this in the broadly commonly accepted way: creation of a BOLD project, an analysis based on K2P parameter and NJ topology, barcoded gap analysis etc to provide a comparability to other studies. Molecular identification does not rely on methods that are used in molecular phylogenetics. As the authors point out, they focus on the identification of larvae: in doing so, they should perform approaches that focus on that. Unfortunately, this is not given yet.

The “broadly commonly accepted way” outlined by the reviewer is actually one of several general approaches; it is not the only acceptable pipeline for DNA barcoding-based specimen identification for many reasons. Modern approaches show much greater diversity in acceptable analysis methods that are influenced by the study system and application (e.g. metabarcoding). Further, the use of NJ topology has been questioned due to known limitations. Thus, maximum likelihood based methods similar to our study are widely considered acceptable (e.g. Doorenweerd et al 2020). Contrary to the reviewer’s assertion, inclusion of a BOLD project is also not required¾ it is widely acceptable to make sequences publicly available on GenBank (e.g. Kim and Jung 2018) where they are periodically “mined” by the BOLD database. Also, as discussed by Etzler et al 2014, in-depth analysis (e.g. barcoding gap, K2P method, phylogeographic patterns) is beyond the scope of studies focused on application of DNA barcoding for identification of morphologically indistinguishable larval stages where sampling is limited (similar to our study). Last, but perhaps most importantly, our methods are directly in line with the most recently published work focusing on gene-based identification in this particular group of insects (Sphenophorus) (Duffy et al 2018).

Still, we do agree that providing analysis that is comparable to other DNA barcoding studies is valuable for the scientific community. Therefore, we have deposited all new sequences generated during this study in GenBank (Lines 181-182) because it is arguably the most widely used public database for sequencing data and, as mentioned earlier, is highly redundant with BOLD. We have also included basic barcoding gap analysis using our adult billbug COI sequences (using BarcodingR program in R – see Lines 201-206) and added a figure showing the distribution of intra- and interspecific K2P distances as requested by the reviewer (Appendix Figure 1A).
